# Mucormycosis in a Severe Trauma Patient Treated with a Combination of Systemic Posaconazole and Topical Amphotericin B—Case Report

**DOI:** 10.3390/antibiotics12101489

**Published:** 2023-09-27

**Authors:** Filip Keller, Helena Antoni, Petra Minarcikova, Ondrej Hrdy, Roman Gal

**Affiliations:** Department of Anesthesiology and Intensive Care Medicine, Faculty of Medicine, Masaryk University and University Hospital, 625 00 Brno, Czech Republic; keller.filip@fnbrno.cz (F.K.); antoni.helena@fnbrno.cz (H.A.); minarcikova.petra@fnbrno.cz (P.M.); gal.roman@fnbrno.cz (R.G.)

**Keywords:** mucormycosis, posaconazole, amphotericin B, severe trauma, case report

## Abstract

Mucormycosis is an opportunistic infection affecting mainly immunocompromised hosts. Infection in immunocompetent patients is rare, but may occur typically in trauma or burn victims. We report on a previously healthy young man suffering devastating trauma from an agricultural accident with the subsequent development of a multifocal mucormycosis. Diagnosis was achieved by cultures obtained from non-healing wounds, some of them even covered by a macroscopic mold formation. Specific treatment was initiated soon after the preliminary results indicated mucormycosis. Aggressive surgical therapy, with concomitant use of systemic posaconazole and topical amphotericin B in a combination treatment, led to the elimination of the fungal infection. The remaining deep tissue defects were consequently reconstructed by a muscle flap and skin graft autotransplantation with a good overall outcome, which would not have been possible without the complete remission of mucormycosis. This case study presents the successful use of a combination treatment with systemic posaconazole and topical amphotericin B and underlines the importance of timely and aggressive surgical therapy.

## 1. Introduction

Mucormycosis is an infectious fungal disease caused by pathogens belonging to the order Mucorales. The most common pathogenic agents are Mucor, Rhizopus and Lichtheimia [1]. The disease is typically opportunistic, affecting mainly immunocompromised hosts. Infection in immunocompetent patients is rare, but can occur in patients suffering from extensive burns or traumatic injuries [2]. Great emphasis is put on timely and radical surgical therapy combined with systemic antimycotic treatment [3]. We report on a case of mucormycosis in a patient with devastating injury caused by agricultural machinery. The main issues complicating the healing process were the sheer extent of the trauma and the subsequent multifocal fungal infection. The treatment choices were also limited by emerging acute kidney injury (AKI).

## 2. Case Presentation

A 24-year-old previously healthy man was admitted to a university hospital in the Czech Republic with a devastating injury to both of his lower extremities, right upper extremity and pelvic and thoracic region (injury severity score 54). The injury was inflicted by agricultural machinery (a drum mower). This device consists of rotating cutting discs capable of vast soft tissue devastation and even bone transection. The patient was intubated on the site of injury and mechanically ventilated; primary hemorrhage control was obtained by the application of tourniquets to both lower extremities on the thigh level and massive volume substitution, before the patient was airlifted to the hospital.

On admission, the patient was sedated and mechanically ventilated, with signs of profound hemorrhagic shock. The initial damage control resuscitation comprised massive hemosubstitution, goal-directed coagulation treatment and pharmacologic support of circulation. A focused assessment with sonography in trauma (FAST) was negative for abdominal free fluid. Immediate damage control surgery was performed, including amputation of the left lower extremity above the knee, the right lower extremity below the knee, external fixation of the right humeral fracture and an initial necrectomy. The patient also suffered from extensive, devastating and heavily contaminated soft tissue injuries (namely, the exposure of vertebrae, ribs, scapula and pelvic bones), which required extensive debridement with the use of a jet lavage system. The patient was provided with initial empiric antibiotic combination therapy consisting of oxacillin, gentamicin and metronidazole. During the initial resuscitation, 11 transfusion units (TU) of red blood cells, 4 TU of fresh frozen plasma, 12 g of fibrinogen concentrate, 1.5 g of tranexamic acid, 4 TU of tromboconcentrate and 2500 IU of protrombin complex concentrate were administered.

A full body CT scan was performed on the second day of hospitalization, revealing the full extent of the trauma (Table 1). Antibiotic therapy was adjusted on day three by the addition of piperacillin/tazobactam and on day five by adding linezolid because of a positive cultivation of *Enterococcus faecium* from a wound swab. A macroscopic blue/green mold formation on the wound in the sacral region was detected during a surgical debridement on day 5, when swabs were obtained.

Preliminary cultivation results from a wound swab received on day seven showed two significant results. Firstly, a nonspecific filamentous fungal growth and, secondly, a positive cultivation of Aspergillus fumigatus, which is susceptible to both voriconazole and posaconazole. Voriconazole infusion therapy had started on the same day with a loading dose of 400 mg twice a day, followed by a dose of 300 mg twice a day.

The condition of the patient was complicated by the development of acute kidney injury (AKI), with subsequent intermittent hemodialysis (iHD) on day 11 and later initiation of continuous veno-venous hemodialysis (CVVHD). Another complication occurred on day 16, in the form of a discrete multifocal intracranial hemorrhage with perifocal edema captured on a CT scan. The use of CVVHD may have contributed to this complication; CVVHD-associated anticoagulation therapy probably led to small-scale intracranial hemorrhages. Inevitable fluid shifts then may have played a role in the development of subsequent perifocal edema. The complication was managed by conservative treatment, with a control CT scan on day 18 showing no progression. Healing was severely limited because of the mycotic infection in the deep tissue defects. Progressive necrosis (most likely due to the known angioinvasive nature of the *Mucor* sp.) demanded extensive surgical debridement. On day 22, Mucor and Rhizopus sp. were cultivated from wound swabs. The antimycotic therapy was then changed from voriconazole, which is ineffective against Mucorales, to posaconazole, with regard to the favorable pharmacokinetic profile of posaconazole in patients with AKI. The posaconazole treatment was initiated with a loading dose, resulting in a total dose of 600 mg during the first day, and then continued with a standard dose of 300 mg per day. Amphotericin B was also applied locally during surgical wound revisions, to potentiate the antimycotic treatment regimen. The wound dressing was soaked in amphotericin B deoxycholate solution (comprising 50 mg of amphotericin B in 1000 mL of sterile water). The dressing was changed every 24–48 h during wound revisions. The surgical revisions and topical therapy were led by a plastic surgery team. The deep wounds required repeated use of a vacuum assisted closure (VAC) device to facilitate healing.

A positive cultivation of Ochrobactrum sp. and a linezolid-resistant strain of Enterococcus faecium from a wound swab on day 23 required a change of antibiotic therapy to tigecycline.

The first skin graft autotransplantation was successfully completed during surgical revision on day 32, as a meshed skin graft from the left thigh was used to cover defects of 10% TBSA on the right thigh and in the left gluteal region.

A sufficient level of consciousness was achieved on day 34 and the patient was freed from a ventilator after a successful spontaneous breathing trial a day later. The deep gluteal defect reconstruction, using a muscle flap supplied by the superior gluteal artery, was successfully completed on day 38. The consequent skin graft autotransplantation of the remaining defects, planned for day 58, was not performed due to the high risk of failure. This was mainly due to unsatisfactory healing and hypercatabolism. The re-alimentation effort was intensified by increasing the daily caloric intake incrementally from 32 kcal/kg/day up to 40 kcal/kg/day, according to the ideal body weight. Recovery was also aided by the administration of nandrolone decanoate on day 88. These interventions, together with ongoing antifungal therapy, enabled a successful skin graft autotransplantation on day 72 and the complete closure of defects in the following weeks.

On day 86, the antibiotic therapy was changed to cefepime. However, four days later, the patient’s consciousness started to deteriorate. This condition required a CT scan of the brain, which showed no changes from previous exams. The suspicion of cefepime neurotoxicity had been raised and was later confirmed by the positive therapeutic effect of iHD.

Finally, on day 99, the patient was transferred to the plastic surgery ward for further reconstructive procedures. The total length of posaconazole therapy was 58 days, with a total of 51 days of combination therapy with amphotericin B administered locally (Figure 1). The intensive care during the first 99 days included mechanical ventilation, CRRT and iHD. There was also a need for surgical tracheostomy, epicystostomy and sigmoidostomy. The mycotic infection and overall problematic healing of the lower extremities required several re-amputations, mainly on the right leg. The final extent of amputations on both lower extremities reached the lower thigh level.

The total hospitalization time (including days spent in the plastic surgery ward) was 165 days. The patient underwent a total of 51 surgical procedures under general anesthesia. The ultimate outcome was positive, the patient´s bowel continuity was restored and he is currently mobile and using prosthetics on both of his lower extremities.

## 3. Discussion

Mucorales are commonly present in soil and vegetation, but manifesting infections are rare because of their opportunistic natures. Mucormycosis is typically divided by its localization into six distinctive types: the rhinocerebral form, pulmonary form, cutaneous form, gastrointestinal form, disseminated form and a group of uncommon forms. Rhinocerebral and pulmonary forms have significantly higher incidence than the others. Rhinocerebral mucormycosis is typically found in immunocompromised patients, often with diabetes or hematologic malignity, and the pulmonary form is frequently associated with induction chemotherapy [4]. Conversely, infection in an immunocompetent patient is rare, but can occur in patients suffering from extensive burns or traumatic injuries. The most frequent form in these individuals is cutaneous or disseminated mucormycosis. The mortality from mucormycosis under the circumstances of severe burns or trauma is exceedingly high; some authors report hospital mortality being as high as 90% [2].

Regarding risk factors, a systematic review focused mainly on burn patients found that infection in two or more locations and also a higher affected total body surface area in burn victims are significantly associated with mortality [3]. The cutaneous form of infection found in our patient is typical for the traumatic mechanism of injury. Our patient was also presented with a major mortality risk factor in the form of multifocal infection (Table 2).

There is a widely accepted consensus on the critical importance of a timely diagnosis and aggressive surgical therapy in patients with mucormycosis [1,2]. Diagnosis is based on histopathological examination and cultures [2]. The diagnosis should not be based solely on histopathological examination, because of its low accuracy in distinguishing Mucorales from other molds, typically Aspergillus sp. There should be verification using culture or molecular identification techniques, if possible [5]. A suitable site for testing may be a non-healing wound, an eschar, or may be even covered by a macroscopic mold formation, which was the case in our patient. Although the invasive character of the mycosis was evidenced by the spread of necrosis between frequent necrectomies, a histopathological confirmation of tissue invasion was not performed in this case. It should also be noted that measurement of the β-D-glucan serum level is not suitable for the diagnosis of mucormycosis, and negative results do not exclude the possibility of this diagnosis [5].

*Mucor* sp. is known for its angioinvasive abilities, which lead to thrombosis and consequently to tissue necrosis. Both of these processes inhibit tissue penetration of antifungals. Angioinvasion can also lead to hematogenous dissemination; therefore, the surgical treatment has to be radical, with the goal of a negative histopathological margin, which requires amputation in many cases. Extensive necrectomy reaching viable tissue then creates favorable conditions for the adequate effect of systemic antifungals in a wound [2]. However, the exact concentration of antifungals in a wound is difficult to determine and, despite adequate surgical therapy, it may not reach the minimal inhibitory concentration (MIC) in some patients [6]. Physicians may attempt to counter this issue by local administration of antifungals, typically amphotericin B. Unfortunately, there are only limited data regarding the effectiveness of this practice [3]. Additional studies regarding the effectiveness of topical amphotericin B in combined therapy with systemic antifungals are needed. However, the rarity of this disease leads to understandable hardships in designing such studies.

Amphotericin B is recognized as a drug of first choice, with limitations in some patient subpopulations. It shows high potency against all members of Mucorales, and the introduction of lipid formulations limits its side effects [3,5,7]. However, even lipid formulations of amphotericin B are still significantly nephrotoxic [8]. According to the European Confederation of Medical Mycology (ECMM) Guidelines 2019, the alternatives for the treatment of mucormycosis are posaconazole and isavuconazole. Both are recommended as salvage therapy in case of refractory mucormycosis or the toxicity of a first-line regimen. They can also be used as a first-line therapy in patients with a high risk of renal impairment [5]. Our patient showed evidence of AKI, thus posaconazole infusion was chosen as the first-line therapy. Amphotericin B was used only topically as an adjunct.

The use of isavuconazole would also be an appropriate choice for a patient with AKI. Isavuconazole, on the contrary to posaconazole, does not contain nephrotoxic cyclodextrin in its IV solution. On the other hand, the VITAL study assessing the safety and efficacy of isavuconazole was criticized for a risk of bias, notably for a small sample size. Nonetheless, these limitations seem to be acceptable when the nature of mucormycosis is considered [9,10].

Data regarding systemic combination therapy for mucormycosis are also scarce. A retrospective study that included 106 hemato-oncological patients, treated for mucormycosis with lipid formulations of amphotericin B and/or posaconazole, found no difference in six-week mortality between the group treated with monotherapy and the group treated with combination therapy [11]. Systemic combination therapy also carries a potential risk of nephrotoxicity enhancement, although it should be stated that there are currently no data proving the possibility of added toxicity.

## 4. Conclusions

The timely diagnosis of mucormycosis and aggressive surgical treatment, paired with effective systemic and topical antifungal therapy, enabled a successful defect reconstruction by a muscle flap with subsequent skin autotransplantation. The combination of systemic posaconazole and topical amphotericin B, administered for almost two months in total, proved to be an effective treatment strategy for our patient, despite multiple negative prognostic factors.

## Figures and Tables

**Figure 1 antibiotics-12-01489-f001:**
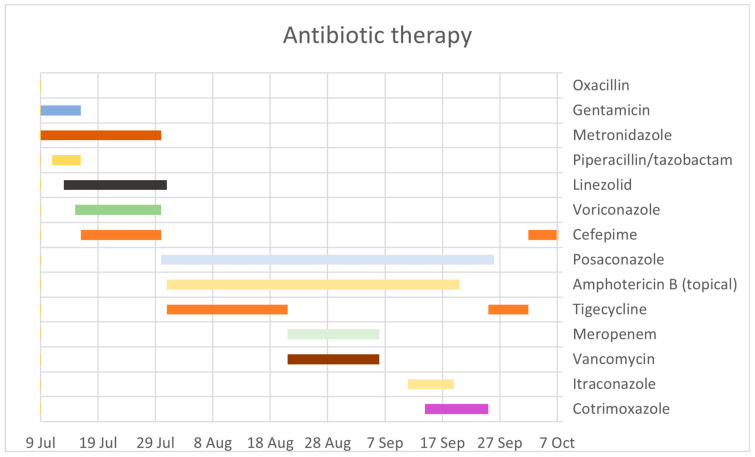
Antibiotic therapy over time.

**Table 1 antibiotics-12-01489-t001:** List of traumatic injuries.

Body Region	Injury
Right upper extremity	Humeral fracture, open
	Deep wounds in scapular region, with complete muscle transection
	Devastating wound in arm region, complete loss of m. triceps brachialis
Left upper extremity	Deep wounds in elbow region
Right lower extremity	Ankle amputation
	Multiple wounds of patella, penetrating into the knee joint
Left lower extremity	Crush injury in crural region
	Large dermal defect in gluteal region
Thorax	Right serial rib fracture (7th to 10th rib), with loss of bone fragments
	Left serial rib fracture (9th to 11th rib)
	Left scapular fracture, open
	Bilateral pneumothorax
	Bilateral hemothorax
	Multiple deep wounds
Pelvis	Right iliac fracture, open, with loss of bone fragments
	Right ischial fracture, open, with dislocation of fragments
	Right pubic fracture, with dislocation of fragments
	Left iliac fracture, with dislocation of fragments
	Devastating wound in right femoral and gluteal region, soft tissue avulsion, complete loss of gluteal muscle, exposure of iliac bone and sacrum

**Table 2 antibiotics-12-01489-t002:** Confirmed cultivation of Mucorales from wound swabs.

*Mucorales* sp.	Body Region
*Mucor* sp.	Right stump
	Left stump
	Left gluteal region
	Lumbar region
*Rhizopus* sp.	Right stump
	Right shoulder
	Left scapula
	Left stump
	Right thigh
	Lumbar region
	Right gluteal region
	Left gluteal region
	Left thigh
*Rhizomucor*	Right thigh
	Right scapula
	Right gluteal region

## Data Availability

The data are available on reasonable request from the corresponding author.

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
