# Peer review of "Mucormycosis in a Severe Trauma Patient Treated with a Combination of Systemic Posaconazole and Topical Amphotericin B—Case Report"

_antibiotics, 2023, doi:10.3390/antibiotics12101489_

Round 1

Reviewer 1 Report

The authors present a very detailed  the management of a case of a patient with severe and complicated trauma. Patient sustained severe injuries and needed multiple and different surgical and other medical interventions.

The author focus of the complication with fungal infection that even grew macroscopic finding in a wound.

They authors discuss what is prevalent in literature, the limitation of how some of those conclusions were drawn al the while relating it to their case. The discuss the therapeutic options in the phase of nephrotoxicity and why they chose a particular antifungal agent. 

Author Response

We would like to thank you sincerely for the kind review.

Reviewer 2 Report

The authors report the case of a young 24 year old male suffering devastating trauma due to agricultural accident. The patient developed multifocal mucormycosis during the course. The authors have given an elaborate step-wise management of the case also taking into consideration the impending AKI. They have discussed various complications encountered in the healing process clearly. Few soft suggestions:

1.       Inclusion of a flowchart depicting various stages in the management and various treatment protocols followed will enhance the quality of paper as well as interest among the readers.

2.       For the antibiotics and systemic antifungal (Posaconazole) administered, were the serum levels of the drugs assessed and any pk-pD correlation done? If yes, please include in text.

3.       There are many spelling and grammatical errors in the text. 

there are many spelling and grammatical errors. 

Author Response

Thank you for your review. We tried to adjust the text according to your implications.

  1. A chart depicting antibacterial and antimycotic treatment was added to the text.
  2. Serum levels of antibiotics were not determined during the therapy, dosing was adjusted empirically based on renal function and renal replacement therapy setting. The patient was closely monitored for clinical signs of drug related toxicity.
  3. We did our best to correct remaining spelling and grammatical errors. Hopefully, it will be enough.

Reviewer 3 Report

Keller and colleagues submit a case report of mucor infection in a trauma patient treated with a combination of systemic and topical therapy.

Major Comment:

You begin to talk about topical Amphotericin B on line 87, but you do not provide any detail. What concentration of Amphotericin did you use? How did you figure out the concentration (did you follow a paper that was published in the literature, and can you reference that in this manuscript)? What was the solution, saline? Did you use a lipid Amphotericin or generic Amphotericin in the solution? Was this a wet-to-dry dressing? How many times a day was the dressing changed (twice)? What was the total length of topical therapy in days?

Minor Comments:

In the second sentence of the introduction, Lichtheimia is spelled wrong and should be corrected.

On line 80, there should be a space between the sentence ending with progression, and the next sentence starting with Healing.

On line 106, nandrolone should be spelled correctly.

On line 170, please separate the words "limited" and "data".

Author Response

We would like to thank you for your review. We tried to adjust the text according to your implications.

Informations about topical therapy were added to the text (lines 90-92).

Major Comment:

Ad 1 (Concentration): The concentration used was 50 mg AmB / 1 L of sterile water (0,005 %)

Ad 2 (Reference): Topical therapy was under the supervision of plastic surgery team, the concentration used was a part of their treatment protocol. Unfortunately, we are not able to trace back the reference.

Ad 3 (Solution): The sterile water was used.

Ad 4 (Type of amphotericin): We used amphotericin B deoxycholate.

Ad 5 (Changing): Dressing with amphotericin B was changed every 24-48 hours during wound revisions.

Ad 6 (Total length of topical therapy): The total length of topical amphotericin B therapy was 51 days. Systemic posaconazole was administered for 58 days.  Both facts are written on lines 117 and 118. The antimycotic therapy length is also depicted in a new chart, which was recently added to the text.

Minor Comments:

Ad 1: Corrected.

Ad 2: Corrected.

Ad 3: Corrected.

Ad 4: Corrected.

Round 2

Reviewer 3 Report

Thank you for allowing me to review your revised manuscript. You have addressed my concerns.